# Optical Coherence Tomography Angiography in Retinal Vascular Disorders

**DOI:** 10.3390/diagnostics13091620

**Published:** 2023-05-03

**Authors:** Charles Jit Teng Ong, Mark Yu Zheng Wong, Kai Xiong Cheong, Jinzhi Zhao, Kelvin Yi Chong Teo, Tien-En Tan

**Affiliations:** 1Singapore Eye Research Institute, Singapore National Eye Centre, Singapore 168751, Singapore; 2Ophthalmology and Visual Sciences Academic Clinical Program (EYE ACP), Duke-NUS Medical School, Singapore 169857, Singapore

**Keywords:** optical coherence tomography angiography, diabetic retinopathy, retinal vein occlusion, retinal artery occlusion, retinal vascular diseases, imaging modality, artificial intelligence, narrative review

## Abstract

Traditionally, abnormalities of the retinal vasculature and perfusion in retinal vascular disorders, such as diabetic retinopathy and retinal vascular occlusions, have been visualized with dye-based fluorescein angiography (FA). Optical coherence tomography angiography (OCTA) is a newer, alternative modality for imaging the retinal vasculature, which has some advantages over FA, such as its dye-free, non-invasive nature, and depth resolution. The depth resolution of OCTA allows for characterization of the retinal microvasculature in distinct anatomic layers, and commercial OCTA platforms also provide automated quantitative vascular and perfusion metrics. Quantitative and qualitative OCTA analysis in various retinal vascular disorders has facilitated the detection of pre-clinical vascular changes, greater understanding of known clinical signs, and the development of imaging biomarkers to prognosticate and guide treatment. With further technological improvements, such as a greater field of view and better image quality processing algorithms, it is likely that OCTA will play an integral role in the study and management of retinal vascular disorders. Artificial intelligence methods—in particular, deep learning—show promise in refining the insights to be gained from the use of OCTA in retinal vascular disorders. This review aims to summarize the current literature on this imaging modality in relation to common retinal vascular disorders.

## 1. Introduction

Optical coherence tomography angiography (OCTA) is a novel, non-invasive, dye-free imaging modality that provides high resolution, depth-resolved imaging of the retinal vasculature in vivo [1,2,3]. OCTA technology is based on detecting signal differences between repeated transverse section optical coherence tomography (OCT) images taken from the same location of the retina in quick succession. These temporal signal differences between the repeated scans are attributed to the movement of erythrocytes within blood vessels, in contrast to the steady signals from static structural background tissue [4,5]. This process highlights areas of blood flow, enabling visualization of the microvasculature down to the capillary level, at a resolution approaching histological precision [6,7]. Furthermore, because OCTA is based on OCT technology, which provides structural cross sectional data of the retina, this allows for blood vessels in different layers of the retina to be segmented and separately analyzed. Based on specific reference planes, various tissue layers or “slabs” can be segmented to produce 2D en face views of different vascular plexuses in a particular layer of the retina (Figure 1A–C) [8]. This is in contrast to traditional fluorescein angiogram (FA) images (Figure 1D), which capture blood flow and vessels as an en face image aggregated from the entire thickness of the retina without the ability to study blood flow within individual retinal layers [4,5]. Novel approaches to OCTA image analysis have increasingly adopted volume rendering to visualize the full 3D vasculature instead of the classic two-dimensional (2D) projection approach [9,10]. Table 1 compares the technical specifications of three commercially available OCTA platforms.

Clinically, OCTA enables the characterization of a wide variety of vascular pathological features, including micro- and macro-aneurysms [11,12,13], capillary remodeling [6], neovascularization [14,15,16], macular telangiectasia [10,17], retinal non-perfusion [18], and venous malformations [15,19]. Consequently, OCTA has opened up a new paradigm in the diagnostic imaging of retinal vascular diseases that will deepen our understanding of these conditions and may ultimately result in clinical benefit for patients. In this paper, we discuss current advances and limitations in OCTA technology and review the application of OCTA to major retinal vascular diseases, including diabetic retinopathy (DR) [15,20], retinal vein occlusions (RVOs), and retinal artery occlusions (RAOs) [21,22].

## 2. OCTA versus Traditional Angiography

Traditional, dye-based angiographic methods, FA and indocyanine green angiography (ICGA), have been considered as the “gold standard” clinical imaging modalities for visualizing vascular changes in the retina and choroid, respectively [23]. However, these methods are subject to several important limitations. First, FA and ICGA require intravenous dye injection, which is invasive, time-consuming, has a limited “transit window”, and a potential adverse side effect profile [24,25]. Second, even under ideal circumstances, these methods have limited resolution and sensitivity [26]. Importantly, FA does not provide detailed resolution down to the level of the retinal capillaries. Third, these methods offer limited depth information due to the 2D, en face nature of image acquisition. Imaging acquired on FA is limited mainly to the superficial vascular plexus, while the deeper retinal capillary plexuses cannot be well appreciated [27]. ICGA enables visualization of the choroidal vessels below the retinal pigment epithelium (RPE) due to its longer wavelength of fluorescence, but still difficulties remain in discerning the different layers of choroidal vessels. Finally, FA exhibits prominent dye leakage, which can be both a strength and a limitation of the modality. Leakage of fluorescein dye provides an assessment of vascular permeability and the integrity of the inner blood-retinal barrier, which is not possible with current OCTA technology. However, excessive dye leakage blurs vessel boundaries and obscures vascular details, particularly in the later frames of the angiogram [27]. Without obscuration by diffuse hyperfluorescence from dye leakage, OCTA is able to generate high-contrast, well-defined images of microvasculature approaching histological resolution [6,26].

There are further practical advantages to OCTA. OCTA acquisition can often be performed on currently available OCT platforms, and because intravenous access and dye administration are not required, OCTA can usually be acquired more quickly than dye angiography. This is especially so, as operators do not need to wait for late frames when acquiring OCTA images. The non-invasive nature of OCTA, coupled with its convenience and quicker acquisition time, allows this test to be repeated more frequently, lending practicality for more regular, interval assessment of retinal diseases, and potentially more longitudinal information as well [1,3]. Detailed comparisons between OCTA and FA/ICGA have been covered extensively in previous reviews [5].

## 3. Limitations and Artifacts in OCTA Imaging

Despite some clear advantages over traditional dye-based angiography, it must also be noted that OCTA comes with unique limitations as well. First, while the lack of dye leakage can be an advantage in terms of visualization and resolution, it is also a key limitation, in that OCTA does not provide information on vascular integrity, permeability or leakage, which are readily and frequently assessed on FA. Second, because of how OCTA works—by detecting differences in signal as a result of erythrocyte movement—it has lower sensitivity than dye-based techniques in capturing low-flow states. Consequently, vascular lesions with low, turbulent flow, such as sclerosed or partially thrombosed microaneurysms or polypoidal choroidal vasculopathy (PCV) lesions, may often go undetected as they fall below OCTA flow detection thresholds [6,28].

Third, OCTA currently has a much smaller field of view compared to dye-based angiography. Current ultra-widefield (UWF) imaging systems allow for a 110° to 220° field of view with dye-based angiography, and this provides valuable information about vascular abnormalities and perfusion in the far periphery, which can affect assessment and management decisions in retinal vascular disease—including DR, RVOs, retinal vasculitis, and sickle-cell retinopathies [29]. In contrast, typical OCTA fields of view are much smaller, ranging from 3 × 3 mm scans focusing on the perifoveal area, up to about 12 × 12 mm scans (Figure 2A), or 15 × 9 mm scans (Figure 2B) achievable with image montage, which capture just beyond the posterior pole and arcades [30,31,32,33,34,35]. As at least two repeat scans for each area are required for motion signal detection, OCTA requires more acquisition time than simple structural OCT scans and under a reasonable acquisition time for patient comfort; OCTA systems, therefore, face an inherent trade-off between field of view and sampling density [36]. Nevertheless, field of view with OCTA is expected to improve with further technological development. Newer, faster, swept source (SS) OCTA scanning systems [37] enable the acquisition of progressively larger fields of view, especially when coupled with extended field imaging (EFI) or montage techniques [38,39,40,41].

Fourth, because of depth resolution and segmentation, the interpretation and analysis of OCTA images is more complex than traditional angiography and have a significant learning curve for both imaging technicians and clinicians. Accurate and meaningful interpretation of abnormalities in each OCTA “slab” requires an appreciation of the segmentation used to produce that particular “slab” image, as well as significant clinical experience and acumen. Errors in segmentation need to be taken into account, and these are particularly common in pathological eyes, such as diabetic macular edema (DME) or age-related macular degeneration (AMD), where there is often disruption of normal anatomic landmarks. Manual adjustment of segmentation boundaries is often required in these cases to obtain good en face images, but this can be tedious and time-consuming.

Fifth, there are multiple different commercially available OCTA imaging platforms that each use different scan protocols, proprietary signal-processing, segmentation algorithms, and different viewing software [27]. This means that there is a learning curve specific to each OCTA platform for clinicians, and more importantly, that OCTA scans and metrics are not comparable across different systems. Incompatibility across different commercial platforms is a major barrier to widespread adoption of clinically useful OCTA biomarkers and quantitative metrics.

Finally, OCTA scans are subject to more artifacts and image quality issues than FA and ICGA [42,43]. Interpretation of OCTA must thus be conducted with a good working knowledge of these possible artifacts. Sources of OCTA imaging artifacts include eye motion, image acquisition, image processing, and display strategies [43,44]. The need for repeated successive B-scans in the same exact location makes OCTA particularly susceptible to motion artifacts, in the form of both bulk motion and saccadic eye movement [45]. Generally, while OCTA algorithms compensate for bulk tissue motion within B-scans [46], saccadic movement poses an imminent challenge. Two solutions are currently in implementation: (1) registering multiple independent orthogonal volume scans and combining the information in post-processing [47]; and (2) incorporating real-time eye tracking technology with OCTA scans [48,49].

Projection artifacts occur when near-infrared light beams passing through superficial blood vessels encounter deeper tissues below. When these light beams are reflected off deeper structures, they may be detected by the OCTA instrument as motion signals in the same pattern as the overlying blood vessels. As a result, the presence of spurious flow (a false or ghost blood flow signal) may be inaccurately identified within deeper layers. Projection artifacts were a major confounding factor in the early inception of OCTA, but modern post-processing algorithms focusing on projection artifact removals have since largely addressed this limitation [44,50,51].

In contrast to projection artifacts, which introduce a false flow signal, shadowing artifacts remove true flow signals and stem from OCT signal attenuation. This is often due to retinal pathology such as drusen, hyper-reflective foci, hard exudates, or extra-retinal features such as floaters or vitreous hemorrhage [51]. These shadowing artifacts may mimic the appearance of non-perfusion areas and capillary dropout [52]. Lastly, an important source of artifacts stems from segmentation error [43]. The parameters used by automated algorithms to segment retinal layers for en face imaging are based on normal retinal architecture and may be especially disrupted in pathological states, including intraretinal fluid, large pigment epithelial defects, or choroidal neovascularization [5]. These lead to segmentation failures creating en face “slabs” which are a merger of disparate vascular layers and can therefore lead to the spurious appearance of abnormal flow or lack of flow. Manual correction of such errors is cumbersome and requires clinical acumen, though the development of interactive segmentation tools which propagate manual adjustments of just a few B-scans over the entire volume may offer a promising and efficient solution to this problem [53].

## 4. Standardization of Reporting of OCTA Findings

With the widespread use of OCTA, it is not surprising that there is some variation and inconsistency in the research literature with regard to image acquisition protocols, segmentation, and reporting of findings. For example, in examining the parafoveal microvascular anatomy of healthy volunteers using seven different OCTA devices, Corvi et al. [54] found that there were significant differences in the parameters measured across OCTA devices, even when the same algorithm and segmentations were used. The authors concluded that the differences in measurements between OCTA devices were beyond a clinically acceptable range. These variations are a major barrier to being able to meaningfully compare between different studies and to the development of clinical useful biomarkers or thresholds that are consistent, reproducible, and relevant.

Particularly in retinal vascular disorders, it is crucial to assess for the extent of ischaemia or non-perfusion, as well as the presence and extent of neovascularization. It is imperative that these findings are reported consistently and accurately, as they have important clinical and research implications. As an example, the extent of capillary non-perfusion would be important to differentiate between ischaemic and non-ischaemic RVOs, and to prognosticate the risk of neovascularization in individual patients. In the existing literature, there is significant heterogeneity in terms used to describe reduced flow signals [55,56,57,58] or capillary non-perfusion [59,60,61].

These issues with consistency and the need for a standardized approach towards OCTA analysis and reporting have been well-recognized, and there are international collaborative efforts to address them. Working towards the standardization of OCTA nomenclature in retinal vascular diseases, Munk et al. conducted first an online survey of retinal specialists [62] and subsequently convened a panel of international experts [63]. From their consensus discussions, they reported that retinal specialists are largely in agreement that OCTA should be used in grading the severity of DR, identifying diabetic macular ischaemia, and differentiating ischaemic from non-ischaemic RVOs [62]. However, things become less certain when it comes to defining specific parameters with potential clinical implications. The experts concluded in their report that a ≥30% change in absolute imaged flow area can constitute a “meaningful change” or a “large area” of reduced flow signal [63]. However, the clinical impact of this cut-off threshold has not yet been clinically validated at this point. Longitudinal studies are needed to evaluate and validate this potential prognostic threshold. Similarly, when it came to defining a “wide-field” OCTA, the experts agree that a field of view greater than 90° would amount to a “wide-field” OCTA [63]. However, after further discussion on whether a field of view was the most ideal method to define a “wide-field” OCTA, the panel of experts were unable to provide a final recommendation. Clearly, these efforts at standardization in OCTA are crucial for the field to progress, and these international collaborative efforts are an important work in progress.

For now, while OCTA has many technical advantages over dye angiography and shows tremendous promise for non-invasive evaluation of the retinal vasculature and prognostication, its clinical applicability is limited by the lack of large-scale longitudinal studies that correlate imaging findings with meaningful clinical outcomes. As this data becomes available, we will be able to develop standardized, validated OCTA thresholds for prognostication in retinal vascular diseases that will be clinically relevant and that will impact clinical decision making.

## 5. Diabetic Retinopathy (DR)

DR is a potentially blinding microvascular complication that affects one-third of patients with diabetes [64]. About 6% of patients with diabetes, representing almost 30 million individuals worldwide, have vision-threatening DR, and these are the eyes at highest risk of blindness [65]. Eyes with DR show various vascular abnormalities that are well-appreciated on OCTA, such as microaneurysms, venous beading and loops, and intra-retinal microvascular abnormalities (IRMAs) (Table 2 and Table 3). The depth-resolved nature of OCTA also allows more comprehensive assessment of these known DR clinical signs and is useful also in evaluating the presence of neovascularization (Figure 2B) [66]. Besides qualitative abnormalities, a major advantage of OCTA is that it can provide quantitative vascular metrics for the assessment and evaluation of DR (Table 3). These include quantitative foveal avascular zone (FAZ) parameters [67] and metrics related to non-perfusion and capillary vessel density [68], which are all being studied for their prognostic significance in DR (Table 2). Furthermore, with OCTA, investigators have been able to detect microcirculatory changes in eyes without clinically-visible DR, which alters our understanding of the early stages of DR or diabetic retinal disease, and may even allow for the evaluation of preventive treatments for early stage disease [69].

### 5.1. Pre-Clinical DR on OCTA

It has long been theorized that microvascular damage from hyperglycemia and subtle alterations in retinal blood flow are the pathophysiologic mechanisms that lead to the clinical signs of DR, such as microaneurysms and retinal hemorrhages [83]. By providing detailed examination of the retinal capillary microvasculature, OCTA has the potential to deepen our understanding of these retinal microvasculature alterations that precede clinically apparent retinopathy.

Cao et al. [84] and Dimitrova et al. [85] both showed reductions in vessel density in the superficial capillary plexus (SCP) and deep capillary plexus (DCP) in diabetic eyes without DR compared to healthy controls. In studies of type 1 diabetics without DR compared to healthy controls, Carnevali et al. [86] and Simonett et al. [87] demonstrated significant reductions in vessel density in the DCP, but not the SCP. This is consistent with most studies showing that the DCP tends to be more severely affected in DR compared to the SCP [88]. However, these OCTA abnormalities in eyes with pre-clinical DR have not been consistently reported across studies. Nesper et al. [89] and Dai et al. [90] both did not find any significant differences when comparing vessel density of the SCP and DCP between diabetics without DR and healthy controls. These mixed findings could be attributed to the use of different OCTA machines with different segmentation algorithms.

Similarly, FAZ abnormalities on OCTA have been reported in patients with diabetes but no clinically visible DR, although these have not always been consistently demonstrated. Dimitrova et al. [85] and Takase et al. [91] found the FAZ to be enlarged in eyes of diabetic patients without diabetic retinopathy compared to that of controls; however, Nesper et al., [89] Dai et al., [90] and Conti et al. [92] did not find this difference in FAZ size.

Histological studies have demonstrated that choriocapillaris abnormalities occur in diabetic eyes without retinopathy [93]. Using OCTA, Choi et al. [94] described choriocapillaris alterations in diabetic eyes without retinopathy. Dai et al., [90] using SS-OCTA to compare flow deficits, found choriocapillaris perfusion in the macula to be reduced in diabetic eyes without retinopathy. However, this also has to be interpreted in light of other studies [85,92] that were not able to demonstrate similar findings.

### 5.2. Clinical DR Lesions on OCTA

OCTA allows for more detailed and accurate evaluation of known clinical features of DR, such as MAs, IRMAs, and NVs.

Ishibazawa et al. [71] correlated MAs detected on clinical examination and fluorescein angiography (FA) with OCTA imaging findings. With the ability of OCTA to resolve depth, Ishibazawa et al. [71] found that MAs were mainly located in the deep plexus. Thompson et al. [70] showed that OCTA allowed for detection of MAs in patients that would otherwise have been undetected on clinical examination. There is interest in the comparison of the ability of OCTA and FA to detect MAs. Schwartz et al. [95] and Ishibazawa et al. [71] showed that there may not be complete agreement between the two modalities in imaging MAs. OCTA may not demonstrate all the MAs that are evident on FA [96]. It has been postulated that this may be because MA detection by OCTA depends on turbulence of blood flow within the MA [97]. However, such limitations may be circumvented by advancements such as multiple en face OCTA image averaging [11].

OCTA is particularly useful for differentiating between IRMAs and NVs in clinical practice. Neovascularization is defined as abnormal vessels that grow through the inner limiting membrane and into the vitreous cavity, and this is the cardinal feature of proliferative diabetic retinopathy (PDR) [98]. With depth resolution and the ability to detect flow, OCTA is ideal for demonstrating neovascular membranes and tufts above the ILM, and therefore able to reliably distinguish them from IRMAs [72,99]. Traditionally, FA has been considered the gold standard for diagnosing NVs in PDR through the demonstration on abnormal leakage [100]. However, there is an argument to be made that OCTA is a better modality to define NVs, as it is able to unequivocally demonstrate NVs anterior to the ILM. Furthermore, there are other advantages in using OCTA for this purpose, such as the ability to further characterize the structural morphology of NVs—as Pan et al. [72] did in proposing three different subtypes of NVs, which may have important prognostic implications. Field of view limitations are also being overcome with the advent of widefield OCTA, with reports showing higher detection rates of PDR on OCTA compared to clinical exam [101].

### 5.3. Diabetic Macular Ischemia on OCTA

Diabetic macula ischaemia (DMI) is an important, but probably underdiagnosed, cause of visual impairment in DR. DMI is traditionally diagnosed with FA as the gold standard [102]. However, it is in the evaluation of DMI that the depth resolution of OCTA becomes particularly important, for instance, in allowing the deep and superficial capillary plexuses to be independently imaged. Novel OCTA metrics of perfusion, such as adjusted flow index [89], have also been developed to add to the assessment of DMI. OCTA therefore allows for better characterization of DMI that was previously not possible with FA. Cheung et al., [103] using OCTA, proposed different clinical phenotypes of DMI—generalized DMI, predominant DCP ischaemia, and predominant SCP ischaemia—that they postulate could explain variability in extent of visual loss between individuals with DMI. As OCTA becomes more widely used in evaluating DMI, it may lead to a deeper understanding that could allow for the development of effective therapies for the prevention or treatment of DMI.

### 5.4. Non-Perfusion in OCTA

OCTA also has an important role in defining areas of retinal non-perfusion [20]. Identifying and quantifying areas of non-perfusion in DR are crucial, as they are an important risk factor for progression to PDR, and for progression of DR severity in general [74,104]. To date, the relationship between retinal non-perfusion and DR progression has been demonstrated mostly on longitudinal cohorts with FA and UWF FA. It is presumed that non-perfusion, as demonstrated on OCTA, would have similar prognostic implications, but this still needs to be demonstrated and validated in prospective longitudinal studies. In fact, some studies have suggested that OCTA may be more reliable for quantification of retinal non-perfusion than FA. Couturier et al. [39] showed in a series of eyes that areas of definite non-perfusion on OCTA were missed on the corresponding UWF FA images due to changes in choroidal background fluorescence. On the other hand, Pellegrini et al. suggested that OCTA, in some instances, may over-diagnose non-perfusion due to segmentation errors, imaging artifacts, or the inability to differentiate between slow-flow and no-flow within a vessel [105]. Nevertheless, OCTA technology continues to improve, and advancements, such as image averaging [106], may reduce the amount of image quality degradation with a wider field OCTA. Methods have also been described to control for artifacts and improve reproducibility of wide-field OCTA [107]. Finally, besides the prognostic implications of detecting non-perfusion on OCTA, it is possible that this may also help to guide treatment. It has been suggested that OCTA may be capable of guiding retinal laser photocoagulation therapy to ischaemic areas in DR, though this approach would need to be validated in prospective clinical trials [108].

### 5.5. OCTA Changes following Treatment in DR

Intravitreal anti-vascular endothelial growth factor (anti-VEGF) injections are used to treat DME, PDR, and sometimes non-proliferative DR. Regular anti-VEGF treatment has been shown to result in regression of DR lesions and reduction in DR severity [109,110]. A key question, therefore, is to determine if anti-VEGF therapy has any effect on areas of retinal non-perfusion, and, in particular, to see if these microvascular changes can be monitored using OCTA. Hsieh et al. demonstrated reductions in the FAZ area and an increase in perifoveal vessel density (VD) [111] after 3 monthly injections of ranibizumab in patients with DME. However, a few other studies examining eyes before and after an anti-VEGF treatment were not able to demonstrate similar post-treatment changes [112,113,114]. Outside the perifoveal region, Couturier et al. also showed convincingly that an anti-VEGF treatment does not result in any improvement of areas of retinal non-perfusion on both OCTA and UWF FA [39]. This is valuable information from a clinical perspective, as it informs clinicians that eyes with apparent regression in DR severity after an anti-VEGF treatment still have significant areas of retinal non-perfusion, and therefore need to be monitored closely for progression, especially if anti-VEGF therapy is stopped or reduced.

After pan-retinal photocoagulation (PRP), some studies have reported changes in foveal and parafoveal vessel density [115,116]. However, these OCTA microvascular changes after PRP have also not been consistently demonstrated [117]. There may be OCTA evidence to support the hypothesis that there is alteration in the dynamics of capillary blood flow at the macula after PRP. Fawzi et al. [118] found no significant difference in vascular density parameters after PRP, but they did find an increase in adjusted flow index—a surrogate measure for capillary blood flow—in the parafovea that may suggest more effective perfusion of the posterior pole. Faghihi et al. [117] also showed on OCTA that circularity of the FAZ seems to improve after PRP as well. In these and other studies examining OCTA changes pre- and post-treatment, analysis of OCTA images has provided valuable insight into the mechanism and effects of various treatments for DR.

## 6. Retinal Vein Occlusion (RVO)

OCTA has been used to investigate vascular changes after RVO, and various qualitative microvascular changes [119], such as vascular tortuosity, vascular dilations, non-perfused areas, collateral vessels, and neovascularization, have been reported in these eyes (Table 2 and Table 3, Figure 3A,B). Quantitative measures [120], such as vessel density and FAZ size and shape, have also been assessed in RVO (Table 2 and Table 3).

In both branch retinal vein occlusion (BRVO) and central retinal vein occlusion (CRVO), studies have demonstrated a reduction in the vessel density of the foveal and parafoveal capillary plexuses [121,122]. The depth resolution of OCTA has also allowed for analysis of different capillary plexuses, which has shown that the DCP is generally more severely affected than the SCP in eyes with RVO [123,124]. It is theorized that this may be due to the direct communication of the DCP with major veins and the lack of vascular smooth muscles in the DCP that may make it more susceptible compared to the SCP [124]. Similarly, this extends to FAZ findings in RVO—enlargement of the FAZ in the DCP is a frequent finding after RVO [41,125]. Enlargement of the FAZ in the SCP has been less consistently demonstrated [41,122,125].

### 6.1. Correlation between OCTA and FA in RVO

FA is the traditional modality for assessing the retinal vasculature and areas of retinal non-perfusion after RVO. However, OCTA is attractive as a convenient, non-invasive alternative to FA for this indication. Vascular changes after RVO on OCTA have been shown to correspond well with those on FA [126]. Nobre Cardoso et al. showed good agreement of FAZ measurements between OCTA and FFA. Shiraki et al. [22,127] and Kadomoto et al. [128] have also demonstrated that areas of retinal non-perfusion quantified by OCTA agreed well with FA in eyes with BRVO.

Assessment of peripheral retinal perfusion is important in eyes after RVO, and in this regard UWF FA has significant advantages over OCTA in terms of the extent of peripheral retina that can be assessed. Current OCTA technology is limited in terms of the size of its scanning area. However, there is some evidence available suggesting that OCTA changes in the posterior retina correlate well with peripheral retinal non-perfusion on UWF FA, and that OCTA may therefore be a reasonable surrogate for UWF FA. Seknazi et al. [129] showed that OCTA macular vascular density correlated well with peripheral retinal non perfusion on FA. Cavalleri et al. [130] found OCTA macular parameters to be correlated with the ischaemic index of UWF FA. Huang et al. [131] similarly found a non-perfusion area in the peripheral retina to be correlated with a nonflow area at the macula on OCTA. Data from these studies suggests that OCTA, even with its relatively limited field of view, has the potential to risk stratify and identify eyes with RVO that may develop neovascular complications, and who may either benefit from an invasive FA, or from closer monitoring and treatment. Finally, particularly as OCTA technology improves and fields of view increase, WF OCTA may be able to adequately visualize the peripheral vasculature directly as well. Li et al. [132] compared UWF-FA with WF OCTA (24 × 20 mm single capture scans) and found both imaging modalities to be comparable in detecting microvascular abnormalities after RVO, which suggests that this is likely to be a viable approach. Glacet-Bernard et al. [80], comparing WF OCTA montage of 5 12 mm × 12 mm images against UWF FA, also concluded that the presence of non-perfusion on WF OCTA correlated well with that on UWF FA.

### 6.2. Structure-Function Correlation in RVO

Functionally, Kang et al. [133] found significant correlations between OCTA quantitative parameters and visual acuity in eyes with RVO. Larger FAZ in the SCP, and lower parafoveal vessel densities in the SCP and DCP were associated with lower visual acuity. Manabe et al. [134] also demonstrated a structure–function correlation with OCTA, showing that there was reduced retinal sensitivity over non-perfused areas relative to perfused areas in eyes after BRVO. Correlating with features on OCT, Moussa et al. [124] showed that increased central macular thickness, disrupted retinal outer layers, and disorganized retinal inner layers on structural OCT were all associated with greater ischaemia on OCTA.

### 6.3. OCTA Changes following Treatment in RVO

Sellam et al. examined a series of eyes with RVO and cystoid macula edema (CMO) that were being treated with anti-VEGF injections [135] and found that anti-VEGF treatment was associated with improvements in qualitative measures of the capillary plexus, such as a decrease in vascular dilation and perifoveal capillary arcade recovery. However, this was also accompanied by a reduction in vessel density that may signify progression of non-perfusion over time. Similarly, Suzuki et al. found that eyes with RVO and CMO showed enlargement of the FAZ 6 months after anti-VEGF therapy [41]. More longitudinal studies with OCTA in eyes receiving treatment for CMO related to RVO will shed greater light on the effects of treatment, and the causes of long-term visual impairment in these eyes.

## 7. Retinal Artery Occlusion (RAO)

Retinal artery occlusions occur as the result of blockage of the central retinal artery (central retinal artery occlusion, CRAO) or its branches (branch retinal artery occlusion, BRAO), which then lead to acute retinal ischaemia and visual loss [136]. OCTA is not necessary for the diagnosis of a RAO, which is made on clinical grounds with OCT imaging. However, it does help to shed light on the pathophysiology of RAOs and the exact levels of retinal ischemia and non-perfusion. OCTA analysis of RAOs in both the acute and chronic phases has demonstrated capillary non-perfusion in affected areas, involving both the SCP and DCP (Table 2 and Table 3). Bonini Filho et al. [21] demonstrated reduced vascular perfusion in SCP and DCP on OCTA that corresponded to areas of delayed perfusion on FA and areas of inner retinal changes on OCT. Shuai et al. [137] similarly demonstrated an ischaemic appearance of both the DCP and SCP that correlated with ischaemic FA changes. Without the choroidal flush on FA that can sometimes make interpretation of capillary non-perfusion difficult, it is qualitatively easier to discern retinal capillary non perfusion on OCTA [138]. Quantitatively, it was also found that vessel density parameters in the SCP and DCP of RAO eyes were reduced compared to fellow eyes and that of normal controls. While the size of FAZ was not found to be enlarged in RAO eyes, the acircularity index (AI), a measure of irregularity of the FAZ, was found to be greater in eyes with RAO [137]. In BRAO, the vessel density of the affected hemifield was also found to be reduced compared to the corresponding unaffected hemifield [137].

There are some limitations to the use of OCTA in RAO. In acute RAO, inner retinal edema may affect interpretation of the perfusion of deeper plexuses due to signal attenuation or projection artifacts [21]. In chronic RAO, there may be segmentation failures as a result of disrupted retinal architecture, loss of definition between inner retinal layers, and significant retinal atrophy [21]. Fixation might also be an issue for RAO eyes as the resultant visual loss is usually significant [138]. As a consequence, image quality has been an issue noted in OCTA studies of RAO [139].

## 8. Artificial Intelligence in OCTA

Alongside clinician-based evaluation of OCTA data, artificial intelligence (AI) approaches have increasingly been applied to take advantage of the large quantities of detailed information available from the 3D, high-resolution, volumetric OCTA scans [52,140,141]. These algorithms seek to extract information that may not otherwise be apparent or accessible from clinical interpretation of OCTA scans alone.

Many of the AI applications for automated OCTA analysis have used deep learning (DL) models. In contrast to the manually-defined feature-based approach adopted by OCTA analytics and clinician interpretations, many DL models forgo feature quantification and aim to detect pathology and disease directly from OCTA slabs or volumetric scans [52,141]. AI models have been developed for the automated detection, prediction, and quantification of pathology from OCTA scans. Proof-of-concept algorithms have been able to use OCTA scans for DR detection and severity staging [142,143,144], RVO detection [145], as well as the automated extraction of specific clinical features such as FAZ evaluation [146], and the detection of non-perfusion/ischaemic areas [147,148].

However, beyond just detection and diagnosis, AI has also been applied to specifically address some of the major limitations in OCTA technology. For example, the time-versus-area restrictions imposed by OCTA motion contrast algorithms may be addressed by AI. Liu et al. and Jiang et al. developed DL models which converted standard static OCT scans into OCTA scans—distinguishing flow and static voxels using image statistics rather than differences across repeated scans [149,150]. Studies by Gao et al. and Kadomoto et al. similarly sought to address the field of view limitations of OCTA by employing DL models to improve the image quality of wide-field but low-definition OCTA scans [151,152]. Other AI models have also been utilized for artefact removal [153,154] and to improve automated slab segmentation algorithms [147,155,156].

With the growing adoption of OCTA in clinical practice and the corresponding increase in the availability of high-quality OCTA data, the role of AI-based approaches in OCTA is primed for improvement. Nonetheless, many current studies remain solely in the “proof-of-concept” stage, and so far, training sample sizes of existing models have been relatively small due to the limited availability of large OCTA data sets [157]. Federated learning paradigms, which enable the collaborative training of AI algorithms across different hospitals without direct exchange of patient data, may offer a promising future avenue to address this current limitation [158,159]. Lastly, external validation and more robust trials are warranted to provide more insight into the true clinical impact of these DL models and on the feasibility of incorporation into clinical practice.

## 9. Conclusions

OCTA has been a major breakthrough in the non-invasive assessment of the retinal microvasculature in health and disease. For retinal vascular disorders in particular, OCTA has been especially useful for examining the differential involvement of the various anatomical capillary plexuses and for providing quantitative vascular metrics that have improved our ability to objectively evaluate retinal vascular pathology. OCTA studies have brought new dimensions to our understanding of major retinal vascular diseases, including DR, RVOs, and RAOs, and have provided better ways to explore structure–function correlations in these diseases.

Nevertheless, there are still a few key barriers to more widespread clinical adoption of OCTA and to direct translation of OCTA data to guide clinical practice and management. Further work to address these limitations will greatly enhance the role of OCTA in clinical practice. First, the development of clear, consistent, and widely accepted OCTA nomenclature frameworks will allow for more standardization in the field and enable more cross-platform compatibility. Second, technology advancements in image acquisition, field of view, image processing, segmentation, and AI will significantly improve OCTA scan quality, accuracy, and field of view. Third, more widespread routine clinical use of OCTA will generate an abundance of additional imaging datapoints that enable the development of better and more powerful AI and deep learning models. There is still yet more potential for OCTA as an imaging modality in retinal vascular disorders to surpass the insights that can be currently gleaned from it.

## Figures and Tables

**Figure 1 diagnostics-13-01620-f001:**
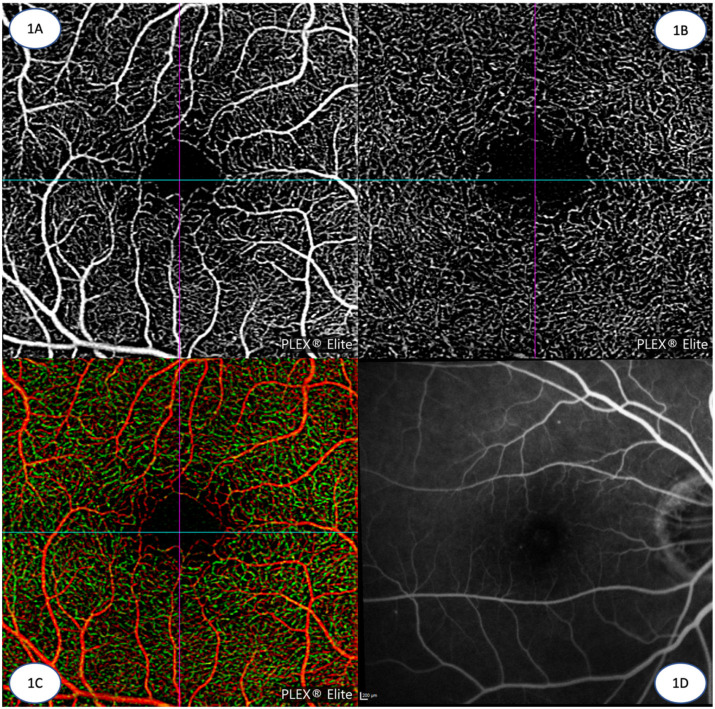
(**A**) 3 mm × 3 mm OCTA image of the superficial capillary plexus in the eye of a healthy control (Zeiss PLEX Elite 9000, Carl Zeiss Meditec, Jena, Germany). (**B**): 3 mm × 3 mm OCTA image of the deep capillary plexus in the eye of a healthy control (Zeiss PLEX Elite 9000, Carl Zeiss Meditec). (**C**): 3 mm × 3 mm OCTA image with color coding combining both the deep (in green) and superficial capillary plexuses (in red) in the eye of a healthy control (Zeiss Plex ELITE 9000, Carl Zeiss Meditec). (**D**): 30 degree fundus fluorescein angiography image of the posterior pole (Spectralis HRA + OCT, Heidelberg Engineering, Heidelberg, Germany).

**Figure 2 diagnostics-13-01620-f002:**
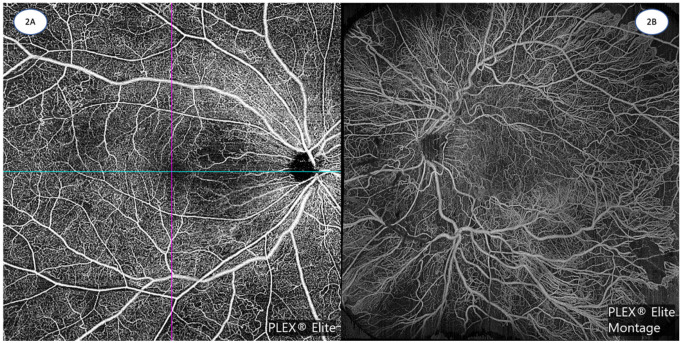
(**A**): 12 mm × 12 mm OCTA image of non-proliferative diabetic retinopathy, demonstrating patchy areas of capillary non-perfusion, microaneurysms, dilated vessels, intraretinal microvascular abnormalities, and mild FAZ enlargement and irregularity. (Zeiss PLEX Elite 9000, Carl Zeiss Meditec). (**B**): 15 mm × 9 mm montage OCTA image of proliferative diabetic retinopathy, demonstrating very large fronds of neovascularization, adjacent to areas of retinal non-perfusion. (Zeiss PLEX Elite 9000, Carl Zeiss Meditec). Images courtesy of K Sandhanam.

**Figure 3 diagnostics-13-01620-f003:**
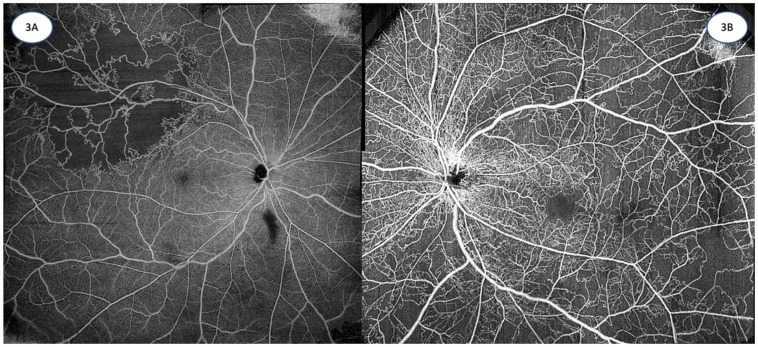
(**A**): 12 mm × 12 mm OCTA image of a superotemporal branch retinal vein occlusion, demonstrating significant areas of retinal non-perfusion, with associated telangiectasias and collateral vessels. (Zeiss PLEX Elite 9000, Carl Zeiss Meditec). (**B**): 15 mm × 9 mm OCTA montage image of a central retinal vein occlusion, depicting extensive areas of capillary non-perfusion with collateral and telangiectatic vessels. (Zeiss PLEX Elite 9000, Carl Zeiss Meditec). Images courtesy of K Sandhanam.

**Table 1 diagnostics-13-01620-t001:** Comparison of the technical specifications of three commercially available OCTA platforms.

OCTA System	AngioVue	SS OCT Angio	AngioPlex
**Manufacturer**	Optovue, Fremont, California, USA	Topcon Corporation, Tokyo, Japan	Carl Zeiss Meditec, Dublin, California, USA
**OCT platform**	RTVue XR AVANTI SD-OCT	DRI Triton SS-OCT	Cirrus 6000 SD-OCT
**Light source**	840 nm	1050 nm	840 nm
**Algorithm**	Split-spectrum amplitude-decorrelation angiography (SSADA)	OCTA-Ratio Analysis (full spectrum amplitude)	Optical microangiography (OMAG)
**Scanning speed**	70,000 scans/s	100,000 scans/s	100,000 scans/s
**Scan area**	3 × 3, 6 × 6, 8 × 8 mm	3 × 3, 4.5 × 4.5, 6 × 6, 9 × 9 mm	3 × 3, 6 × 6, 8 × 8, 12 × 12, 14 × 10 (montage), 14 × 14 (montage) mm
**Axial optical resolution**	3 microns	8 microns	5 microns
**Lateral optical resolution**	15 microns	20 microns	15 microns
**Axial imaging depth**	2.0–3.0 mm	2.6 mm	2.9 mm

**Table 2 diagnostics-13-01620-t002:** Qualitative and quantitative OCTA abnormalities in diabetic retinopathy and retinal vein occlusion.

Diabetic Retinopathy OCTA Abnormalities	
Qualitative	Microaneurysms [70,71] Intraretinal microvascular abnormalities (IRMAs) [72] Venous beading and loops [73] Neovascularization at the disc (NVD) and elsewhere (NVE) [15,72]
Quantitative	Increased areas of capillary non-perfusion [74] Reduced SCP and DCP vessel density [18,75] Reduced fractal dimension [75] Increased FAZ size [75] Reduced FAZ circularity [76] Reduced SCP and DCP vessel length density [77] Reduced vascular area density [77] Wider venular calibre [77]
**Retinal Vein Occlusion OCTA Abnormalities**	
Qualitative [78,79]	Microaneurysms Vessel tortuosity and dilatation Telangiectatic vessels Collateral vessels Optic disc collaterals Neovascularization
Quantitative	Increased areas of capillary non-perfusion, [80] Reduced SCP and DCP vessel density [79,81] Reduced fractal dimension [82] Increased FAZ size [79,81] Reduced FAZ circularity [81] Increased lacunarity [82]

**Table 3 diagnostics-13-01620-t003:** Summary of key OCTA insights based on disease entity.

Disease Entity	Key OCTA Insights
Diabetic Retinopathy	Provides quantitative, depth resolved vascular and foveal avascular zone parameters that allow assessment of different capillary plexuses Derivation of novel metrics from vascular parameters Detection of microvasculature abnormalities before the onset of visible clinical signs Allows further quantification and characterization of diabetic macular ischaemia as well as known diabetic retinopathy signs Allows better differentiation between intra-retinal microvascular abnormalities and neovascularisation Quantification of retinal non-perfusion that could potentially guide treatment and may be more reliable than fluorescein angiogram Determine impact of treatment on perfusion status
Retinal Vein Occlusion	Provides quantitative, depth resolved vascular and foveal avascular zone parameters that allow assessment of different capillary plexuses Derivation of novel metrics from vascular parameters Quantification of retinal non-perfusion that may guide further management Determine impact of treatment on perfusion status
Retinal Artery Occlusion	Provides quantitative, depth resolved vascular and foveal avascular zone parameter that allows assessment of different capillary plexuses Derivation of novel metrics from vascular parameters

## Data Availability

Data sharing not applicable. No new data were created or analyzed in this study. Data sharing is not applicable to this article.

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
