# Peer review of "Optical Coherence Tomography Angiography in Retinal Vascular Disorders"

_diagnostics, 2023, doi:10.3390/diagnostics13091620_

Round 1

Reviewer 1 Report

This paper presents a permonorized study on the advantages and disadvantages of using the new imaging modality of the retinal vasculature, Optical Coherence Tomography Angiography (OCTA), for non-invasive evaluation. Through a literature review, the authors present the OCTA studies and what they have brought new to the understanding of major retinal vascular diseases, including RD, RVOs, and RAOs, and provide better ways to explore structure-function correlations in these illnesses. However, through the elaborate review, the authors still present the difficulties of OCTA for a wider clinical adoption and for the direct translation of its data in clinical practice.

The article is very well written and organized, playing an important role in its research area. The elaborated study is quite complete and significant. A detailed review identifying the positive and negative aspects of using OCTA.

Just a few suggestions that can improve the paper:

- The hyphenation of words throughout the text.

- Sentences with lots of whitespace between them, for example on lines: 116, 126, 195, 261, 287, 295, 328, 332, 399.

- In table 1, commas are missing in the separation at the end of some lines in column 2.

- The formatting of the legends in figure 2 and 3 must be changed in the same way as the legend in figure 1.

- The formatting of chapter 9 of the conclusions must be the same as the formatting of the remaining chapters.

The paper is well written and does not require major changes in English.

Author Response

29 April 2023

Editor, Diagnostics

RE: Ong et al. Optical Coherence Tomography Angiography in Retinal Vascular Disorders.

Dear Editor Prof. Dr. Andreas Kjaer, MD, PhD, DMSc,

Thank you for the opportunity to revise our manuscript and address the reviewers’ comments. We have responded to all concerns and revised the manuscript to incorporate the required changes.

We managed to improve the manuscript through the process of responding to the reviewers’ comments. Please find attached below the comments of the reviewers along with our responses and changes in the manuscript.

Please do not hesitate to contact us if you have any further questions or require further clarifications or modifications to the manuscript.

We look forward to working with you on this manuscript.

Sincerely,

Dr Tien-En Tan, MBBS(Hons), MMed(Ophth), FRCOphth, FAMS

Associate Consultant, Singapore National Eye Centre, Singapore

Singapore Eye Research Institute, Singapore

Duke-NUS Medical School, Singapore

11 Third Hospital Avenue, Singapore 168751

Phone: (65) 65767200

Email: tantienen@gmail.com

Reviewer 1

Reviewer’s Comments

Author Response and Changes Made

This paper presents a permonorized study on the advantages and disadvantages of using the new imaging modality of the retinal vasculature, Optical Coherence Tomography Angiography (OCTA), for non-invasive evaluation. Through a literature review, the authors present the OCTA studies and what they have brought new to the understanding of major retinal vascular diseases, including RD, RVOs, and RAOs, and provide better ways to explore structure-function correlations in these illnesses. However, through the elaborate review, the authors still present the difficulties of OCTA for a wider clinical adoption and for the direct translation of its data in clinical practice.

The article is very well written and organized, playing an important role in its research area. The elaborated study is quite complete and significant. A detailed review identifying the positive and negative aspects of using OCTA.

Thank you for your comments.

We have made specific changes below to address your comments and hope that you will find the manuscript improved.

Just a few suggestions that can improve the paper:

- The hyphenation of words throughout the text

Thank you for pointing this out.

We have reviewed the hyphenation of words throughout the text and edited the manuscript.  

- Sentences with lots of whitespace between them, for example on lines: 116, 126, 195, 261, 287, 295, 328, 332, 399.

Thank you for pointing this out.

We have removed the unnecessary whitespaces.

- In table 1, commas are missing in the separation at the end of some lines in column 2.

Thank you for pointing this out.

We have edited table 1 accordingly.

- The formatting of the legends in figure 2 and 3 must be changed in the same way as the legend in figure 1.

Thank you for pointing this out.

We have edited the formatting of the legends of figures 2 and 3 so that there is uniformity throughout the paper.

- The formatting of chapter 9 of the conclusions must be the same as the formatting of the remaining chapters.

Thank you for pointing this out.

We have edited the formatting of chapter 9 to be consistent with the remaining chapters.

Reviewer 2 Report

This is a well written review about OCTA, with interesting insights and very useful to the sector.

It could be interesting to add some technical insights, and the optical scheme of the device, the acquisition times etc and to compare them among the most diffused commercial providers of this product.

Moreover, a comprensive table highliting the state of advancements of the technique as a diagnostic tool in the several pathologies discussed could provide a useful summary of authors'finding.

The English is good

Author Response

29 April 2023

Editor, Diagnostics

RE: Ong et al. Optical Coherence Tomography Angiography in Retinal Vascular Disorders.

Dear Editor Prof. Dr. Andreas Kjaer, MD, PhD, DMSc,

Thank you for the opportunity to revise our manuscript and address the reviewers’ comments. We have responded to all concerns and revised the manuscript to incorporate the required changes.

We managed to improve the manuscript through the process of responding to the reviewers’ comments. Please find attached below the comments of the reviewers along with our responses and changes in the manuscript.

Please do not hesitate to contact us if you have any further questions or require further clarifications or modifications to the manuscript.

We look forward to working with you on this manuscript.

Sincerely,

Dr Tien-En Tan, MBBS(Hons), MMed(Ophth), FRCOphth, FAMS

Associate Consultant, Singapore National Eye Centre, Singapore

Singapore Eye Research Institute, Singapore

Duke-NUS Medical School, Singapore

11 Third Hospital Avenue, Singapore 168751

Phone: (65) 65767200

Email: tantienen@gmail.com

Reviewer 2

Reviewer’s Comments

Author Response and Changes Made

This is a well written review about OCTA, with interesting insights and very useful to the sector.

Thank you for your comments.

We have made specific changes below to address your comments and hope that you will find the manuscript improved.

It could be interesting to add some technical insights, and the optical scheme of the device, the acquisition times etc and to compare them among the most diffused commercial providers of this product.

Thank you for pointing this out.

We agree that it would indeed be interesting to have further technical details of the devices included and have hence drawn up Table 1 in the revised manuscript that has technical specifications of three commercially available OCTA platforms.

Moreover, a comprensive table highliting the state of advancements of the technique as a diagnostic tool in the several pathologies discussed could provide a useful summary of authors'finding.

Thank you for this comment.

We have drawn up Table 3 in the revised manuscript that summarises key OCTA insights based on disease entity and hope this highlights the state of advancement of these diagnostic tools.